# The missing Northern European winter cooling response to Arctic sea ice loss

James A. Screen[1]

Reductions in Arctic sea ice may promote the negative phase of the North Atlantic Oscillation (NAO − ). It has been argued that NAO-related variability can be used an as analogue to predict the effects of Arctic sea ice loss on mid-latitude weather. As NAO − events are associated with colder winters over Northern Europe, a negatively shifted NAO has been proposed as a dynamical pathway for Arctic sea ice loss to cause Northern European cooling. This study uses large-ensemble atmospheric simulations with prescribed ocean surface conditions to examine how seasonal-scale NAO − events are affected by Arctic sea ice loss. Despite an intensification of NAO − events, reflected by more prevalent easterly flow, sea ice loss does not lead to Northern European winter cooling and daily cold extremes actually decrease. The dynamical cooling from the changed NAO is 'missing', because it is offset (or exceeded) by a thermodynamical effect owing to advection of warmer air masses.

---

[1] College of Engineering, Mathematics and Physical Sciences, University of Exeter, 603 Laver Building, North Park Road, Exeter EX4 4QE, UK. Correspondence and requests for materials should be addressed to J.A.S. (email: J.Screen@exeter.ac.uk).

The Intergovernmental Panel on Climate Change Fifth Assessment report[1] found that warming of the climate system is unequivocal and human influence on the climate system is clear. The rapid retreat of Arctic sea ice cover is one of the most visible manifestations of man-made climate change[2–4]. The annual minimum sea ice cover (in September) has declined by 40% from 1979 to 2015 and is now lower than that at any other time in the past 1,450 years[5]. Climate model simulations run with increasing greenhouse gas concentrations unanimously project continued loss of sea ice, with ice-free summers the norm later this century if greenhouse gas concentrations continue to rise[6–9]. This profound environmental change has motivated extensive research aimed at understanding the climatic implications of sea ice loss, both within the Arctic and beyond[10–15].

The response of the large-scale Northern Hemisphere atmospheric circulation to Arctic sea ice loss has proven hard to elucidate, owing to its inherent nonlinearity[15–17]—with respect to the magnitude and spatial pattern of sea ice loss[18–21] and to the background climatic state[22,23]—apparent model dependence[24], and often low detectability amidst the large chaotic variability of the system[25]. Despite this large uncertainty, a common conclusion is that reductions in Arctic sea ice tend to favour a shift towards the negative phase of the NAO[26], or its hemispheric equivalent, the Arctic Oscillation (AO). A causal link between Arctic sea ice and the NAO (or AO) has been inferred from observations/reanalyses[27–36], seasonal predictions[37] and climate model simulations[19,20,24,38–46]. Although such a negative shift of the NAO has been found in many studies, there are exceptions[17,21,25,47,48], for reasons that are not well understood.

The negative phase of the NAO is associated with cooler winter temperatures over Europe[26]. Therefore, it has been assumed (often implicitly or via association) that sea ice loss will favour colder winters over Europe (and mid-latitudes more generally) if, as evidence suggests, sea ice loss promotes the negative NAO phase[49–52]. However, it is plausible that the European winter temperature response to Arctic sea ice loss is influenced by factors other than the negative NAO shift. Furthermore, although several studies have suggested a physical link between Arctic sea ice loss and winter cooling over Asia[18,53–58], connections to European winter climate are less clear[14]. Extreme caution is required when extrapolating conclusions from one mid-latitude region to another.

This study presents evidence from model simulations that strongly support the notion of a negative NAO response to Arctic sea ice loss. This atmospheric circulation change would be expected to lead to cooling over Europe, if the NAO is a good analogue for the expected temperature response to Arctic sea ice loss. However, such a cooling response is 'missing' in these model simulations, because it is offset (or exceeded) by a thermodynamical effect owing to advection of warmer air masses.

## Results

**Sea ice loss.** This study makes use of large-ensemble atmospheric model simulations with perturbed sea ice conditions to isolate the influence of Arctic sea ice loss on the negative phase of the NAO (NAO−). It focuses on the NAO− for two key reasons. First, the climatological winter mid-tropospheric circulation response to sea ice loss in these simulations projects onto the NAO− (Supplementary Fig. 1), prompting a closer look at NAO− events specifically. Second, considering the wider literature, the one dynamical change that appears commonplace (if not ubiquitous) in response to sea ice loss is a tendency towards NAO−. The main analyses are based upon two 502-member ensembles, one with below-average sea ice cover and the other with above-average sea ice cover (see Methods for further details), hereafter referred to as the low ice (LI) and high ice (HI) ensembles. Figure 1a,b show the differences in sea ice concentration between LI and HI during early winter (November–December) and midwinter (January-–February), respectively. There are reduced sea ice concentrations, in LI compared with HI, along the sea ice edge and in the sub-polar seas. The difference patterns are largely similar between early and midwinter, except for larger sea ice reductions in Hudson Bay and the Chukchi Sea in the former, and larger reductions in the Sea of Okhotsk and Labrador in the latter. The Barents–Kara Sea is a region where sea ice reductions are understood to be especially effective at influencing the NAO[18,33,36,46,56]. Decreased ice cover in this region is evident in both early and midwinter.

Figure 1c,d show differences in sea ice concentration between two additional experiments, referred to as the twenty-first century (C21) and twentieth century (C20) ensembles (see Methods). The differences in sea ice cover between C21 and C20 are more spatially extensive than between LI and HI, and the difference in sea ice area is roughly twice as large (− 8.9 versus − 4.8 million km$^2$ in early winter; − 10.2 versus − 5.2 million km$^2$ in midwinter). However, in some regions, the differences in sea ice concentration are larger between LI and HI than between C21 and C20 (Supplementary Fig. 2). This is especially the case near to the observed climatological sea ice edge (stemming from the fact that C20 has less sea ice than HI). The C21 and C20 ensembles will be utilized later, but first the focus is on the differences in the NAO between LI and HI.

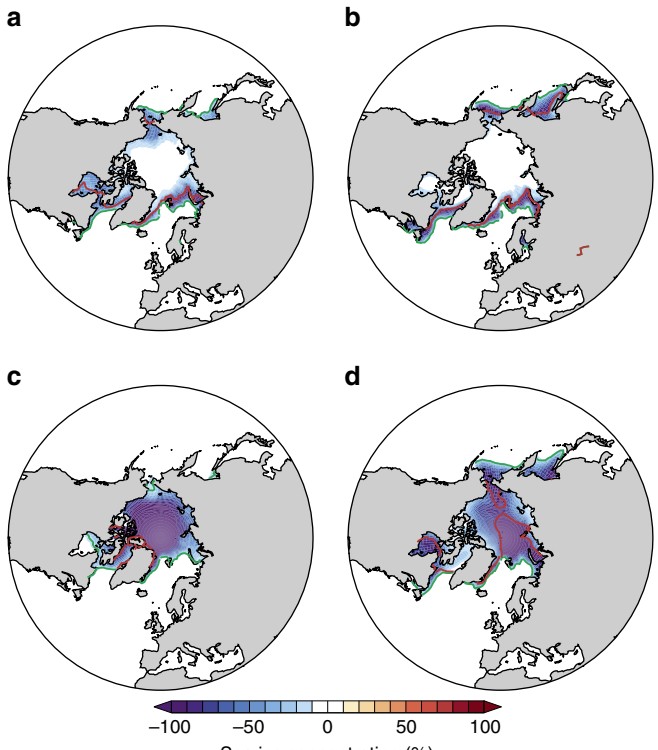

**Figure 1 | Arctic sea ice loss prescribed in the model simulations.**
(**a**) Early winter (November–December) sea ice concentration differenced between the low ice (LI) and high ice (HI) simulations (LI minus HI). (**b**) As **a** but for midwinter (January–February). (**c,d**) As **a,b** but for the difference between twenty-first century (C21) and twentieth century (C20) simulations (C21 minus C20). The red and green lines mark the sea ice edge (15% contour) in LI and HI, respectively, in both **a,b** and in C21 and C20, respectively, in both **c,d**.

**NAO response**. Unlike previous studies that have examined seasonal mean changes of the NAO in response to Arctic sea ice loss, the large ensembles used here allow in-depth analysis of strongly negative NAO events specifically. This distinction is important as changes in extreme events (in this case, low surface NAO index values) may not simply follow changes in mean climate and society is arguably more sensitive to the former. This study focuses on seasonal-scale (January–February mean) NAO− events, which are associated with prolonged periods of anomalous weather and significant impacts on society. Simulated NAO− events (defined here as when the midwinter surface NAO index is >1 s.d. below its mean; see Methods) are characterized by raised mid-tropospheric (500 hPa) geopotential heights centred over Greenland and lowered heights over the North Atlantic (Fig. 2a, contours). These anomalies are vertically coherent and extend from the surface into the stratosphere, as illustrated by a vertical cross-section along the 40° W meridian (Fig. 2b, contours).

To estimate the influence of Arctic sea ice loss on NAO− events, the difference is taken between a composite mean of NAO− events in LI and that in HI. Mid-tropospheric height

differences between NAO− events in LI and HI (Fig. 2a, shading) project strongly onto the climatological NAO− pattern (Fig. 2a, contours). NAO− events are associated with raised heights over Greenland and depressed heights over the North Atlantic in LI compared with HI. The vertical profile of height differences between NAO− events in LI and HI (Fig. 2b, shading) also closely resembles the vertical structure of the climatological NAO− (Fig. 2b, contours). These differences imply that midwinter NAO− events are amplified (intensified) by Arctic sea ice loss. This intensification can also be seen as significant ($P < 0.001$) increase in the s.d. of the surface NAO index (1.36 hPa (95% confidence intervals: 0.86 to 1.86)) but no significant ($P = 0.96$) change in its mean (0.02 hPa ($-0.64$ to 0.68); Supplementary Fig. 3). It is noteworthy that the climatological midwinter circulation response is NAO like in the mid-troposphere but not at the surface, hence no mean shift in the surface NAO index. This study focuses on midwinter (January–February) as the intensification of NAO− events is most pronounced in these months (Supplementary Fig. 4); however, other studies have found NAO responses to be maximal in late winter[19,20]. The timing of the NAO response may be dependent of the atmospheric model used and/or the sea ice conditions prescribed.

**Troposphere–stratosphere interaction**. The temporal evolution of polar cap (>65° N) height (PCH) is a commonly used metric to infer the evolution of the NAO (or AO) through time[19,20,46]. Figure 2c shows the evolution of PCH in the months preceding, during and following midwinter NAO− events, and how this differs between LI and HI. Typically, midwinter NAO− events are preceded by increases in stratospheric PCH in late autumn and early winter months (Fig. 2c, contours). These positive PCH anomalies descend through time and become apparent in the troposphere by midwinter. Although the stratospheric PCH anomalies persist into early spring following the midwinter NAO− event, the tropospheric anomalies dissipate. Comparing LI and HI, PCH is enhanced in the stratosphere from October to March in LI with positive tropospheric anomalies emerging a few months later in December and persisting until April. This familiar response pattern to Arctic sea ice loss[19,20,46] strongly suggests a warming and weakening of the stratospheric vortex, followed by a downward propagation of circulation anomalies into the troposphere with a lag of around 1–2 months. Negative PCH differences in the stratosphere during spring may be linked to delayed final breakdown of the polar vortex, which often follows the recovery from a weakened winter vortex.

As large differences in stratospheric PCH emerge in November preceding midwinter NAO− events (Fig. 2c), attention now turns to this month and potential causes of the weakened polar stratospheric vortex. Previous work has suggested that sea ice loss increases vertical wave propagation into the stratosphere in early winter and leads to a weakened polar vortex[46]. Such increases in vertical wave activity are understood to relate to amplification of the climatological planetary waves, in particular the zonal wavenumber 1 component[46]. The concept of linear interference—how the forced response interacts with the climatological waves—appears a powerful paradigm to explain the effect of extratropical surface forcing, such as sea ice loss, on vertical wave activity[59–61]. Figure 3 shows the zonal wavenumber 1 component of the difference in geopotential height between LI and HI for Novembers preceding midwinter NAO− events. The differences display a westward tilt with altitude, indicative of vertical wave propagation[62], and are tightly in phase with the climatological wavenumber 1. Thus, these simulations support the notion that the planetary wave response to Arctic sea ice loss

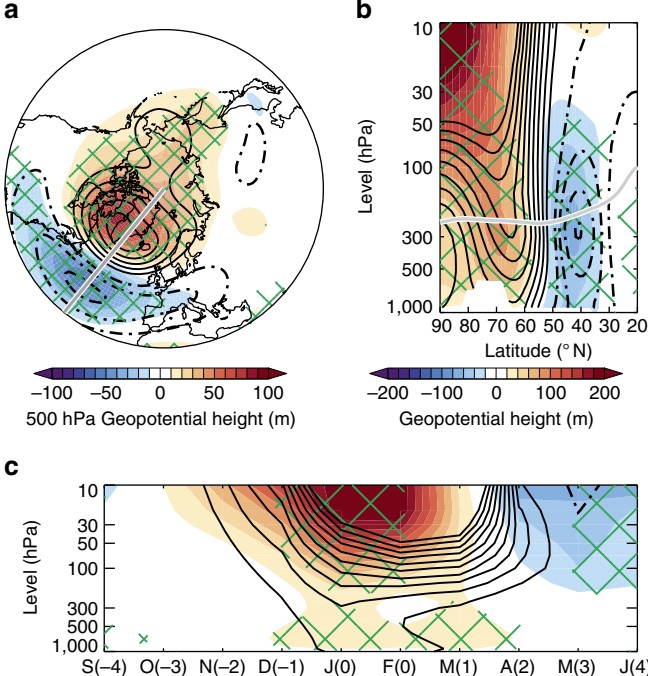

**Figure 2 | Changes to NAO− events induced by Arctic sea ice loss.** (**a**) Midwinter (January–February) 500 hPa geopotential height during NAO− events, differenced between the low ice (LI) and high ice (HI) simulations (shading; LI minus HI). (**b**) Midwinter geopotential height along the 40° W meridian (marked by a grey line in **a**) during NAO− events, differenced between LI and HI (shading). (**c**) Polar cap (north of 65° N) averaged geopotential height for the 4 months preceding (September–December; leftmost), during (January–February; centre) and 4 months following (March–June; rightmost) midwinter NAO− events, differenced between LI and HI (shading). Green hatching (**a–c**) denotes differences that are statistically significant at the 95% ($P = 0.05$) confidence level. Black contours (**a–c**) show the average geopotential height for NAO− events relative to climatology (average of both LI and HI; solid for positive; dashed for negative; drawn from −200 to 200 at intervals of 20 m, excluding zero). The solid grey line in **b** marks the tropopause.

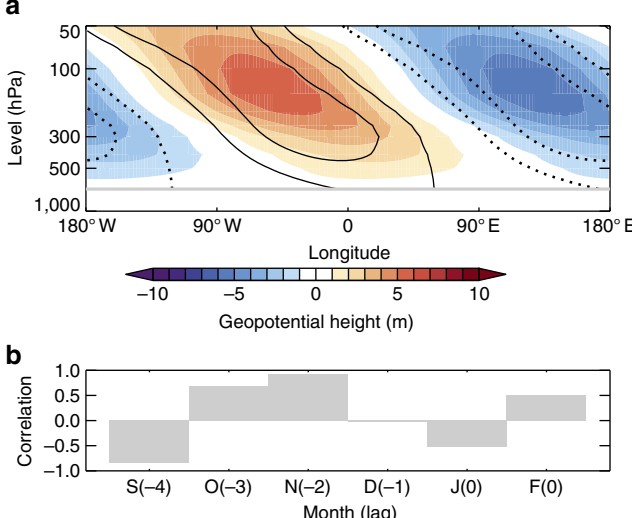

**Figure 3 | Planetary wave interference induced by Arctic sea ice loss.**
(**a**) Zonal wavenumber 1 component of November geopotential height averaged between 40 and 60° N during NAO − events, differenced between the low ice (LI) and high ice (HI) simulations (shading; LI minus HI). The black contours show the climatological wavenumber 1 (average of both LI and HI; solid for positive, dashed for negative; drawn from − 300 to 300 at intervals of 50 m, excluding zero). Values below 700 hPa are omitted due to pressure levels intersecting elevated topography. (**b**) Pattern correlation (50-700 hPa) between the forced wavenumber 1 response and the climatological wave for the months preceding and during midwinter NAO − events.

interferes constructively with the climatological wave pattern in November and enhances vertical wave propagation into the stratosphere, consistent with ref. 46. In other months, the wave response does not project so well onto the climatological wave pattern (Fig. 3b), which suggests enhanced vertical wave propagation in November (and to a lesser extent October) triggered by sea ice loss is especially relevant for winter weather, as proposed by others[33,37,46]. It is worth noting that despite using a 'low-top' model (that is, with a model lid at 10 hPa and relatively poor vertical resolution of the stratosphere) in this study, the results are strongly consistent with those from 'high-top' models[19,20,46]. In summary, these simulations display a robust intensification of NAO − events in response to Arctic sea ice loss, through a mechanism whereby enhanced tropospheric wave activity leads to a weaker stratospheric polar vortex and precedes more intense NAO − events.

**NAO-related temperature response.** Focus now shifts to the effects of NAO − events on near-surface (1.5 m) temperature. The NAO explains the largest percentage of midwinter temperature variance over a region covering 15° W–40° E 50–65° N (Fig. 4a), encompassing the British Isles, Belgium, the Netherlands, northern Germany and Poland, the Baltic States and southern Scandinavia, and hereafter referred to as Northern Europe. One-third (33.2%) of the simulated variance in midwinter Northern European 1.5 m temperature is explained by the NAO. Northern European temperature is 2.26 °C (1.84 − 2.68) colder than average during NAO − events (Table 1). This cooling is understood to be largely related to easterly wind anomalies and enhanced advection of cold continental air masses into Northern Europe. Averaged over Northern Europe, NAO − events are associated with mean westerlies of 0.61 m s$^{-1}$ compared with an average of 1.98 m s$^{-1}$ (Table 1). Figure 4b shows the spatial

pattern of temperature anomalies during NAO − events. Cooler temperatures also occur over Siberia, East Asia and North America; however, in these regions the NAO explains a smaller fraction of the total variance than over Northern Europe.

One might expect that the temperature difference between NAO − events in LI and HI (Fig. 4c) would resemble an amplified NAO − temperature pattern (Fig. 4b), given the intensification of NAO − events by sea ice loss. However, this is not the case. Over Siberia, NAO − events are associated with warmer temperatures in LI compared with HI (Fig. 4c), rather than cooler temperatures that would be expected from more intense NAO − events. Over most of Europe, there is little change in temperature associated with NAO − events, despite the intensification of these events. Specifically for Northern Europe, there is a marginally significant (P = 0.06) warming despite a highly significant (P < 0.001) decrease in zonal wind (Table 1), the latter implying a more easterly flow regime typically linked to colder winter temperatures. This is called the missing cooling response, referring to the fact that midwinter Northern European temperature is unaffected by sea ice loss despite the marked intensification of NAO − events that would be expected to yield cooling. A lack of Northern European cooling is also apparent in the climatological midwinter response to sea ice loss (that is, including all midwinters not just NAO − ones; Supplementary Fig. 1).

Better understanding of the reasons for this missing cooling response can be obtained by considering the anatomy of NAO − events using simulated daily data. Figure 5a compares histograms of Northern European daily zonal wind for all midwinters and for NAO − events in LI and HI. During NAO − events (in both LI and HI), there are more frequent days of easterly zonal wind compared with climatology and, conversely, fewer days of westerly zonal flow. Comparing NAO − events in LI and HI, there are more easterly days and fewer westerly days in the former than the latter (Fig. 5a). Thus, the reduction in midwinter mean zonal wind over Northern Europe induced by sea ice loss is associated with more days of easterly flow. Comparable histograms for surface temperature (expressed as anomalies from the daily climatology in HI to remove the effects of the seasonal cycle) reveal more frequent days of below-average temperature during NAO − events (in both LI and HI) and, in particular, more frequent occurrences of cold extremes (Fig. 5b). There are notable differences in the histograms of daily temperature during NAO − events between LI and HI, despite the small change in mean temperature. There are fewer occurrences of temperature anomalies lower than − 3 °C in LI compared with HI, but more occurrences of anomalies in the range − 3 to 7 °C (Fig. 5b). In other words, although sea ice loss increases the number of moderate cold and warm anomalies, the largest cold anomalies decrease in number. These opposing differences result in only a small change in mean temperature.

There is a strong linear relationship between Northern European daily zonal wind and temperature during NAO − midwinters, evident in both LI and HI (Fig. 5c). As mentioned earlier, easterlies tend to be associated with colder conditions and vice versa. This linear relationship can be used to estimate the temperature change that one would expect for a given change in zonal wind: the decrease in zonal wind of − 0.58 m s$^{-1}$, between LI and HI, yields an anticipated cooling of 0.50 °C. This contradicts the simulated warming of 0.68 °C (− 0.02 to 1.39).

**Dynamical and thermodynamical effects.** The temperature difference between NAO − events in LI and HI (0.68 °C (− 0.02 to 1.39)) can be partitioned into contributions coming from days of differing daily zonal wind strength (Fig. 5d, black line). Further, it is possible to estimate a contribution, owing to the change in

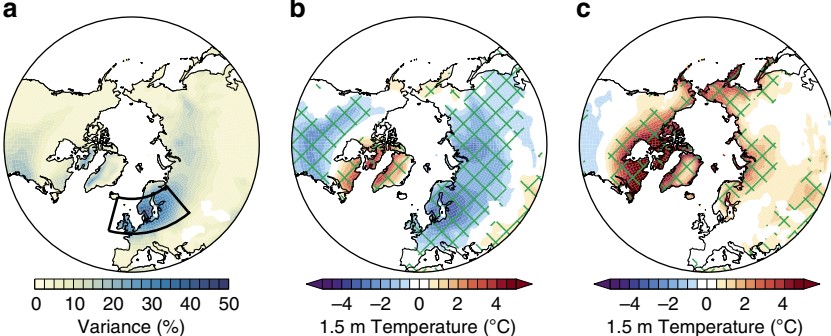

**Figure 4 | Effects of the NAO on near-surface temperature. (a)** Percentage of midwinter (January–February) 1.5 m temperature variance explained by the NAO (average of both low ice (LI) and high ice (HI) simulations). **(b)** Midwinter 1.5 m temperature during NAO − events (average of both LI and HI) relative to climatology. **(c)** Midwinter 1.5 m temperature during NAO − events, differenced between LI and HI (LI minus HI). The black box in **a** marks the Northern European domain. Green hatching (**b**,**c**) denotes differences that are statistically significant at the 95% ($P = 0.05$) confidence level.

**Table 1 | Changes in Northern European winter climate.**

|  | Mean (all) | Mean (NAO −) | Difference (NAO −) |
|---|---|---|---|
| 1.5 m Temperature | − 4.15 | − 6.41 | 0.68 ( − 0.02 to 1.39) |
| 10 m Zonal wind | 1.98 | 0.61 | **− 0.58 ( − 0.88 to − 0.27)** |

HI, high ice; LI, low ice; NAO, North Atlantic Oscillation.
Simulated 1.5 m temperature (°C) and 10 m zonal wind (m s$^{-1}$) over Northern Europe (black box in Fig. 4a): in all midwinters (average of both LI and and HI simulations); during NAO − events (average of both LI and HI); and the difference between NAO − events in LI and HI (LI minus HI). Differences (fourth column) significant at the 95% confidence level (confidence intervals provided in parentheses) are highlighted in bold italic font.

frequency of days in each wind category, assuming no change in the average temperature of days in each wind category, and a contribution owing to a change in the average temperature associated with each wind category, assuming no change in the frequency of days in each wind category. As the former describes a change of wind in the absence of a mean temperature change and the latter, a mean temperature change in the absence of changes in circulation, these are referred to as dynamical and thermodynamical components, respectively. The dynamical component (Fig. 5d, blue line) is dominated by a cooling contribution on days of zonal wind in the range − 5 to 2 m s$^{-1}$. These are more frequent in LI relative to HI and are associated with cold temperature anomalies; hence, they act to lower the midwinter temperature in LI relative to HI. It is noteworthy that although days of zonal wind less than − 4 m s$^{-1}$ are also increased, these are few in number so contribute less to the midwinter mean change. There is a smaller dynamical cooling contribution from days of zonal wind in the range 4–8 m s$^{-1}$, owing to fewer of these typically warmer days in LI compared with HI. The dynamical contribution is small for days of zonal wind in the range 0–3 m s$^{-1}$. In contrast, the thermodynamical contribution (Fig. 5d, red line) is largest in this range, but positive for all categories. As all wind categories are warmer in LI compared with HI (Fig. 3c), the magnitude of the thermodynamical contribution is largely dictated by the mean frequency of each category, with more frequent categories making a larger contribution to the midwinter mean temperature difference. The net contribution (Fig. 5d, black line) shows cooling (dynamically driven) on days of strong easterly flow ( < − 5 m s$^{-1}$) and on days of strong ( > 5 m s$^{-1}$) westerly flow, and warming (thermodynamically driven) on days of moderate ( − 5 to 5 m s$^{-1}$) easterly and westerly flow. Summed over all days in midwinter NAO − events (that is, over all wind categories), the cooling effect of intensified NAO − is missing, owing to a larger and opposite warming effect.

**Cold extremes**. Figure 5b shows a reduction in the frequency of daily cold extremes. This reduction in cold extremes is caused in part by mean warming, but also by decreased daily temperature variability, consistent with previous work[20,57]. Reduced variability is a physical consequence of weakened horizontal temperature gradients[63,64]. Northern European cold extremes tend to be associated with advection of cold subpolar air from northern Eurasia, a region that is warmed by Arctic sea ice loss (Fig. 4c). It is worth noting that cold extremes decrease in frequency (Fig. 5b), despite a net cooling on days of strongest easterlies ( < − 5 m s$^{-1}$; Fig. 5d). This can be understood by the fact that the zonal wind is only one factor of many that influences temperature, meaning that not all the coldest days are coincident with strong easterlies. Some of the coldest days fall into wind categories that are warmed by sea ice loss (for example, − 5 to 5 m s$^{-1}$; Fig. 5d), which explains the reduction in cold extremes.

**Robustness of the response**. Past work has suggested that the atmospheric response to Arctic sea ice loss can be dependent on the magnitude and spatial pattern of sea ice loss[17–21] and the background climatic state[22,23]. Therefore, a pertinent question to ask is: is the missing Northern European cooling response a feature specific to these simulations or a consistent feature of the atmospheric response to Arctic sea ice loss? To begin to explore this question, the large ensemble was sub-sampled into four smaller ensembles corresponding to the four different background states (see Methods). Both the climatological NAO − response (that is, across all midwinters; there are too few NAO − events in each of the smaller ensembles to allow reliable comparison of solely NAO − midwinters) and the absence of Northern European cooling are robustly simulated in all four cases (Supplementary Figs 5 and 6), suggesting little sensitivity to the background state (this is in contrast to other aspects of the response to sea ice loss[23]). To further explore potential sensitivities, it is useful to attempt to reproduce the results using another set of ensemble simulations with the same model, but very different prescribed sea ice concentrations. These additional ensembles were briefly introduced earlier (C21 and C20). Recall that the difference in sea ice area between C21 and C20 is approximately twice as large as between LI and HI, and the spatial patterns of sea ice loss are very different (Fig. 1 and Supplementary Fig. 2). Despite these differences in sea ice forcing (and in the background state), there is very high consistency in the simulated atmospheric response to sea ice loss. Mid-tropospheric heights differenced between C21 and C20, show a very similar amplification of NAO − events (Fig. 6a) to that discussed previously (Fig. 2a), suggesting this is a robust

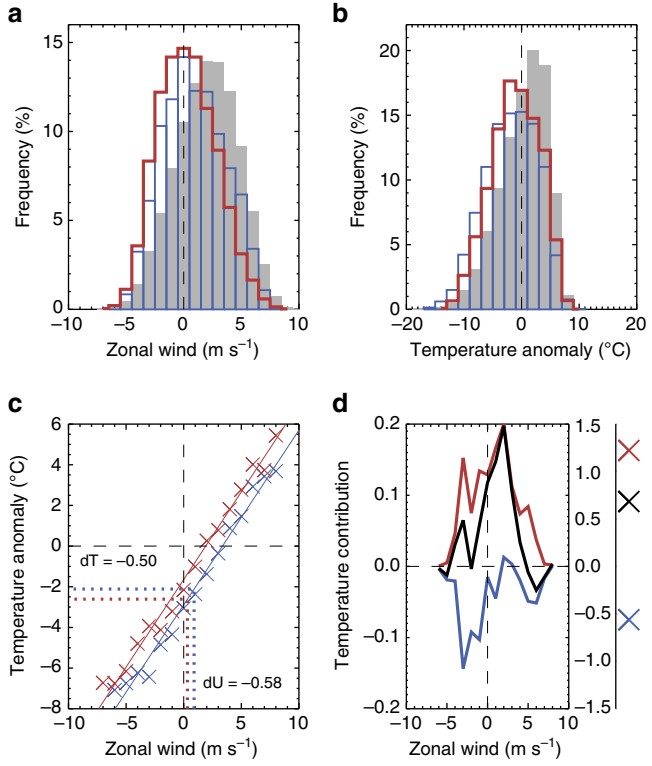

**Figure 5 | Anatomy of NAO− events changed by Arctic sea ice loss.**
(**a**) Histograms of daily 10 m zonal wind averaged over Northern Europe (black box in Fig. 4a) for all midwinters (January–February; grey bars; both low ice (LI) and high ice (HI) simulations) and for NAO− events in LI and HI (red and blue lines, respectively). (**b**) As **a**, but for daily 1.5 m temperature anomalies (relative to the daily climatology in HI). (**c**) Relationship between daily 10 m zonal wind and 1.5 m temperature anomalies during NAO− midwinters. Each cross corresponds to a Northern European and bin average (classified by zonal wind with a bin size of 1 m s⁻¹) in LI (red) and HI (blue). The solid lines show linear relationships, referred to in the main text with the blue line used to predict the expected temperature change (dT) due to the simulated change in zonal wind (dU; LI minus HI). (**d**) Dynamical (blue), thermo-dynamical (red) and net (black) contributions to the midwinter mean difference in Northern European 1.5 m temperature during NAO− events between LI and HI. The line graph shows the contributions as a function of daily 10 m zonal wind and the crosses show the total contribution.

feature of the atmospheric response to sea ice loss (at least in this model). Another consistent feature is the absence of European cooling associated with these more intense NAO− events (c.f. Figs 4c and 6b).

Histograms of Northern European daily zonal wind and temperature during NAO− events reveal very similar changes between C21 and C20 (Fig. 6c,d) to those reported earlier between LI and HI (Fig. 5a,b). Namely, an increase in days of easterly flow, an increase in days of moderate cold anomalies and a decrease in days of large cold anomalies. The difference in midwinter Northern European zonal wind between NAO− events in C21 and C20 is of comparable magnitude (−0.60 m s⁻¹ (−0.95 to −0.26); Fig. 6e) to that between LI and HI (−0.58 m s⁻¹ (−0.88 to −0.27); Fig. 5c), despite much larger sea ice differences between C21 and C20 (Fig. 1 and Supplementary Fig. 2), emphasizing that the dynamical response does not scale linearly with the magnitude of sea ice loss[15–17]. The enhanced easterlies exert a dynamical cooling contribution but, as found before, this is offset by thermodynamical warming (Fig. 6f),

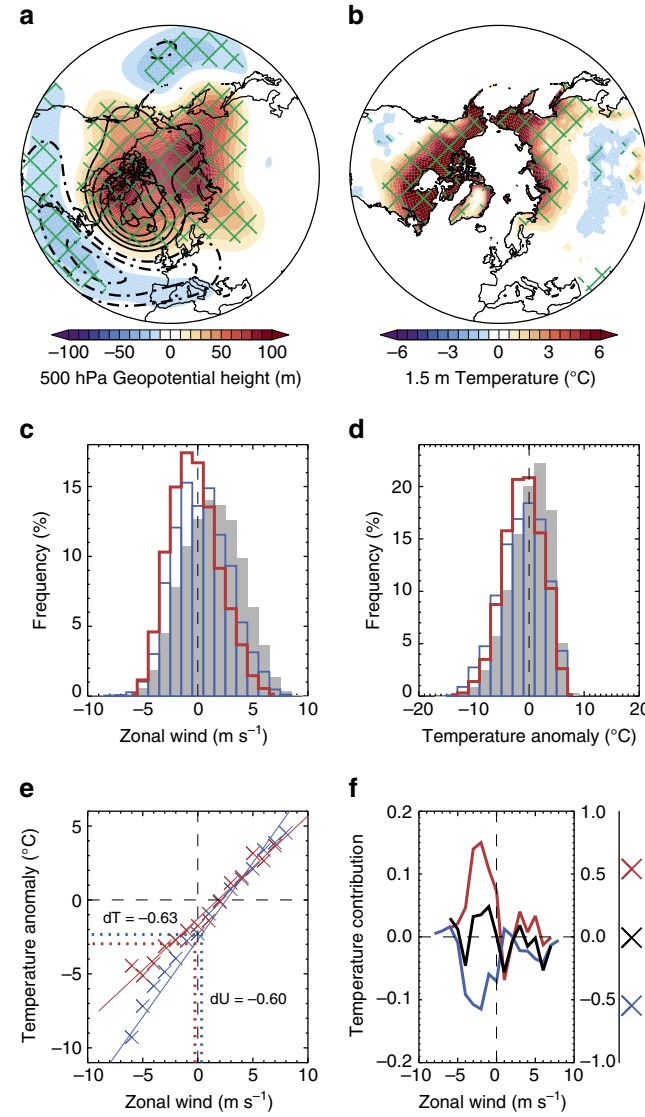

**Figure 6 | Replication of main results in simulations with different Arctic sea ice loss.** (**a**) Midwinter (January–February) 500 hPa geopotential height during NAO− events, differenced between the twenty-first century (C21) and twentieth century (C20) simulations (shading; C21 minus C20). (**b**) As **a**, but for 1.5 m temperature. Green hatching (**a,b**) denotes differences that are statistically significant at the 95% (P = 0.05) confidence level. Black contours (**a**) show the average geopotential height for NAO− events relative to climatology (average of both C21 and C20; solid for positive; dashed for negative; drawn from −200 to 200 at intervals of 20 m, excluding zero). (**c**) Histograms of daily 10 m zonal wind averaged over Northern Europe (black box in Fig. 4a) for all midwinters (January–February; grey bars; both C21 and C20) and for NAO− events in C21 and C20 (red and blue lines, respectively). (**d**) As **c**, but for daily 1.5 m temperature anomalies (relative to the daily climatology in C20). (**e**) Relationship between daily 10 m zonal wind and 1.5 m temperature anomalies during NAO− midwinters. Each cross corresponds to a Northern European and bin average (classified by zonal wind with a bin size of 1 m s⁻¹) in C21 (red) and C20 (blue). The solid lines show linear relationships, referred to in the main text with the blue line used to predict the expected temperature change (dT) due to the simulated change in zonal wind (dU; C21 minus C20). (**f**) Dynamical (blue), thermo-dynamical (red) and net (black) contributions to the midwinter mean difference in Northern European 1.5 m temperature during NAO− events between C21 and C20. The line graph shows the contributions as a function of daily 10 m zonal wind and the crosses show the total contribution.

leading to a small and insignificant ($P = 0.81$) mean temperature response (0.10 °C (−0.66 to 0.87)). Despite an increase in frequency (as well as intensity) of NAO− midwinters in C21 relative to C20 (see Methods), there is no evidence of Northern European cooling in the climatological temperature response to sea ice loss (that is, across all midwinters not just NAO− ones; Supplementary Fig. 7), even though the climatological circulation response is similar to the NAO−.

One noteworthy difference between the two sets of simulations is that the slope of the wind–temperature relationship is reduced in C21 compared with C20 (Fig. 6e), whereas it stays roughly constant in HI and LI (Fig. 5c). This difference in slope is attributable to larger (sub-) polar warming in the future sea ice loss scenario and further weakened horizontal temperature gradients. In summary, the intensification of NAO− events and the missing Northern European cooling are common features of the response to Arctic sea ice loss in both sets of simulations. The absence of NAO-like surface temperature change, despite a circulation response to sea ice loss reminiscent of the NAO−, was also evident in ref. 24 (although it was not explored in any detail). However, both sets of simulations analysed in this study and the simulations analysed in ref. 24 were conducted with the same model, so model dependence cannot be ruled out.

## Discussion

The NAO is a key driver of winter weather and climate variability over Northern Europe. Given the similarities between the mean atmospheric state during NAO− events and that often simulated in response to Arctic sea ice loss (for example, Figs 2a and 6a), the NAO has been suggested as a prototype to understand how mid-latitude weather might change with Arctic sea ice loss. As NAO− winters are typically colder than average, the above line of reasoning would predict that Arctic sea ice loss causes winter cooling over Northern Europe. This study strongly suggests, however, that Northern European winter temperature is only weakly affected by Arctic sea ice loss, despite a marked intensification of NAO− events. The temperature of seasonal-scale NAO− events remains fairly constant (or warms), because thermodynamical warming offsets (or exceeds) NAO-related dynamical cooling. Furthermore, using the NAO as an analogue would predict more frequent cold extremes over Northern Europe, although the simulated response suggests fewer such events. Thus, the NAO− cannot be used as an analogue to predict how surface temperature responds to Arctic sea ice loss. Further work is required to ascertain whether this holds true for modes of atmospheric variability other than the NAO. In this context, it is noteworthy that a similar conclusion was recently drawn in relation to the AO as an analogue for the effect of Arctic warming on atmospheric blocking[65].

This study has shown that a Northern European cooling response is missing in these simulations and explained its absence, but why then are Arctic sea ice reductions correlated with cold winters in the real world[49–52]? The simulations strongly suggest that although Arctic sea ice loss may augment the negative NAO, the European cooling correlated with sea ice loss in observations is not caused by sea ice loss. Instead, it is likely to be related to co-varying atmospheric variability[52,66,67]. In other words, the observed correlation between Arctic sea ice and European winter temperature does not appear to be indicative of a physical relationship. This study has only considered the effects of sea ice loss and it remains to be seen how co-varying factors, such as Eurasian snow cover, may influence connections between sea ice, the NAO and Northern European weather.

Research into linkages between the Arctic and mid-latitudes is in part motivated by the potential to improve prediction of mid-latitude weather[33,68]. The results here suggest that Arctic sea ice cover could be a potential source of predictability for the NAO. Indeed, November sea ice cover in the Kara Sea has been identified as one possible contributor to skilful NAO predictions in a state-of-the-art seasonal prediction system[37]. However, the results also suggest that improved predictions of the NAO may not translate in better forecasts of surface temperature unless the temperature of advected air is also well predicted.

In conclusion, the study provides support for a causal link between Arctic sea ice loss and more intense midwinter NAO− events but, importantly, emphasizes that cooling over Northern Europe stemming from this dynamical change is fully compensated by thermodynamical warming.

## Methods

**Model.** Model simulations were performed with the UK Met Office Unified Model version 6.6.3, which is the atmospheric component of the Coupled Model Intercomparison Project 5 (CMIP5) model HadGEM2-ES[69]. The model was utilized in an atmosphere-only configuration with prescribed surface boundary conditions. The atmosphere-only framework has the distinct advantage that sea ice can be perturbed in a controlled way, to isolate its influence on the atmosphere. The major weakness of this approach, however, is that it fails to capture coupled atmosphere–ocean–ice interactions and feedbacks, which may modify the atmospheric response[44,45,67]. External forcings (for example, greenhouse gas concentrations, aerosols and so on) were held constant. The model version used here has a horizontal resolution of 1.875° longitude and 1.25° latitude (known as N96) and 38 vertical levels.

**Low ice and high ice ensembles.** Two ensemble experiments were performed with either positive or negative sea ice anomalies. Both experiments consist of 502 ensemble members, with each member being 1 year in duration and having the same surface boundary conditions, but starting from a different atmospheric initial condition. For sea ice boundary conditions, the monthly mean climatological mean and s.d. ($\sigma$) of observed sea ice concentration and sea surface temperature (SST), 1979–2013, was calculated at each grid point from the UK Met Office Hadley Centre Ice and SST data set (http://www.metoffice.gov.uk/hadobs/hadisst). In the HI experiments, a sea ice concentration anomaly of $+2\sigma$ was applied to the climatological mean and for the LI experiments an ice concentration anomaly of $-2\sigma$ was applied to the climatological mean. At grid points where a sea ice anomaly was imposed, an SST anomaly was also imposed to account for SST changes linked to sea ice changes, adapting the approach of ref. 24. In the HI experiment, an SST anomaly of $-2\sigma$ was applied to the climatological mean and in LI an SST anomaly of $+2\sigma$ was applied to the climatological mean. At grid points where sea ice is never present or always has the same concentration, the climatological sea ice concentration and SST was used. Specific ice-related anomalies are applied in each calendar month, but only in the northern hemisphere. Sea ice concentrations were restricted to being between 0 and 100%, to avoid unphysical values. Sea ice thickness was calculated empirically within the model code from the prescribed sea ice concentrations. Previous work has shown that the atmospheric response to sea ice loss can be sensitive to the background SST[22,23]. For this reason, four different background states are used, intended to capture SST variability associated with the Pacific Decadal Oscillation (PDO) and the Atlantic Multidecadal Oscillation (AMO). These were chosen, as they are the dominant modes of SST variability on decadal to multi-decadal timescales in the Pacific and Atlantic oceans, respectively, and this study is focused on the response to Arctic sea ice loss on these timescales. To represent the different PDO phases, the detrended and normalized annual-mean PDO index (http://www.esrl.noaa.gov/psd/data/climateindices/list/), 1948–2013, was regressed against detrended annual mean global SST, to yield an SST anomaly per $1\sigma$ change in the PDO index ($\beta$). For one background state, an SST anomaly of $+2\beta$ was applied and the other an SST anomaly of $-2\beta$ was applied. The SST anomalies were applied globally at all ice-free grid points, with the same SST anomalies applied in each calendar month. SSTs were restricted to no lower than $-1.8$ °C (freezing temperature of saltwater) to avoid unphysical values. An analogous procedure was applied for the AMO to yield two additional background states. In both LI and HI (to ensure no net difference in SST between these), the two PDO background states were each applied in 150 members and the two AMO background states were each applied in 101 members.

**Twenty-first and twentieth century ensembles.** Two 260-member ensembles were performed with either sea ice conditions representative of the late twentieth century (C21) or those projected for the late twenty-first century (C20). For the C20 experiment, sea ice concentrations and SSTs were taken from the CMIP5 historical simulations of HadGEM2-ES, averaged for the period 1980–99 and across all available ensemble members ($N = 4$). For the C21 experiment, sea ice concentrations were taken from the CMIP5 RCP8.5 simulations of HadGEM2-ES,

averaged for the period 2080–99 and across all available ensemble members ($N = 4$). SSTs in C21 were the same as C20, except at grid boxes where sea ice was lost, where the climatological SST of the late twenty-first century was used. This procedure accounts for the local SST warming associated with reduced sea ice cover. The RCP8.5 simulations are forced by a continuous increase in greenhouse gas concentrations and are often viewed as a 'business-as-usual' scenario, with limited mitigation strategies applied. This scenario was chosen to maximize the signal-to-noise ratio. Further details on these simulations can be found in ref. 57.

**NAO events and compositing.** NAO indices were calculated from the midwinter (January–February) weighted area-average mean sea level pressure over the domain 0–80° W 30–50° N minus that over the domain 0–80° W 60–80° N. Sensitivity tests confirmed that the results were robust for alternative NAO definitions, for example, based upon the Principal Component time series of the leading Empirical Orthogonal Function of sea level pressure over the Atlantic sector (Supplementary Fig. 8). The NAO indices were normalized by subtracting the ensemble mean and dividing by the ensemble s.d. The mean and s.d. were determined separately for each experiment; however, the results are highly consistent when normalization is relative to the HI experiment (Supplementary Fig. 9). A surface NAO index value of − 1 or lower was classified as an NAO− event, yielding 71 events in both LI and HI, and 49 and 42 events in C21 and C20, respectively. It is noteworthy that the number of events differs between C21 and C20 due to differences in higher-order moments (more negative skewness and kurtosis). The difference in NAO− events induced by Arctic sea ice loss was estimated from by subtracting the composite mean of NAO− cases in HI (C20) from that in LI (C21). A Student's *T*-test was used to assess the statistical significance of the differences, which compares the sample means to the variances within both samples and accounts for unequal variances between samples. The null hypothesis of equal means is rejected with 95% confidence when $P \leq 0.05$.

**Dynamical and thermodynamical roles.** The methodology to decompose the dynamical and thermodynamical contributions was adapted from ref. 70. Rather than classifying each day based on a two-dimensional spatial pattern, here each day was classified based on the strength of 10 m zonal wind averaged over Northern Europe (15° W–40° E 50–65° N), using 15 bins from − 6 to 8 m s$^{-1}$ with an interval of 1 m s$^{-1}$. The dynamical and thermodynamical contributions were estimated from:

$$\Delta T = \sum_{1}^{N} (T_i \Delta f_i + f_i \Delta T_i + \Delta T_i \Delta f_i) \qquad (1)$$

where $\Delta T$ is the total change in temperature between LI (C21) and HI (C20), $T_i$ is the bin-averaged temperature in HI (C20), $f_i$ is the frequency of occurrence of bin i in HI (C20), $\Delta f_i$ is the change in frequency of occurrence for bin i between LI (C21) and HI (C20), $\Delta T_i$ is the change in bin-averaged temperature between LI (C21) and HI (C20), and $N$ is the total number of bins (in this case, $N = 15$). The first term, $T_i \Delta f_i$, relates to changes in the frequency of occurrence of particular wind regimes and provides an estimate of the dynamical contribution. The second term, $f_i \Delta T_i$, relates to changes in temperature averaged over all days that belong in each bin, which provides an estimate of the thermodynamical contribution. The third term, $\Delta T_i \Delta f_i$, represents the contribution from the interaction of both changing wind regime and bin-averaged temperature. It is included in the net contributions shown (for completeness), but is not presented as an individual term, as it was found to be small compared with the other two terms.

**Software.** All graphics were produced using IDL version 8.2.2.

**Data availability.** All relevant data are available from the corresponding author on request.

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

## Acknowledgements

J.A.S. was funded by UK Natural Environment Research Council grants NE/J019585/1, NE/M006123/1 and NE/P006760/1. The model simulations were performed on the ARCHER UK National Supercomputing Service. Observational data sets were provided by the NOAA Earth System Research Laboratory and Met Office Hadley Centre.

## Author contributions

J.A.S. designed and performed experiments, analysed data and wrote the paper.

## Additional information

**Competing financial interests**: The author declares no competing financial interests.

**How to cite this article**: Screen, J. A. The missing Northern European winter cooling response to Arctic sea ice loss. *Nat. Commun.* **8**, 14603 doi: 10.1038/ncomms14603 (2017).

**Publisher's note**: 

