## [Peer Review File · Nature Communications]

Reviewers' comments:

Reviewer #1 (Remarks to the Author):

This paper is a modeling study that explores the atmospheric response to Arctic sea ice loss using a large ensemble of sensitivity numerical experiments. The originality of the study lies in investigating causes for the apparent inconsistency between the dynamical response to sea ice loss (negative NAO-like pattern) and the near-surface temperature response that exhibits little or no winter cooling over Northern Europe. The author shows evidence that the thermodynamical effect of sea ice loss, i.e. a surface warming at the pole and in adjacent regions, offsets the effect of the dynamical response due to advection of relatively warmer air when sea ice is decreased.

The study provides an interesting contribution to the field, that is going to help interpreting the temperature response to Arctic sea-ice loss in other similar numerical studies. Anticipating the effect of Arctic Amplification on the mid-latitude climate is a great challenge for climate scientists, and this study highlights an important aspect, that is that a loss of Arctic sea ice does not necessarily lead to cooler winter over Europe as expected from dynamical considerations. The paper is well-written, but I think some aspects could be clarified, especially concerning the methodology and interpretation of the results. In particular, I'd like to see a discussion of the overall impact of sea ice loss in other winters than NAO- ones, and also clarify whether the conclusion of this study applies to other weather patterns than NAO-. The discussion and main conclusions can be improved too.

Please see hereafter a list of comments. I am confident that the paper will be suitable for publication after improving these points, I therefore recommend a major revision.

Major comments

- My first major comment concerns the fact that the study focuses on the impact of sea ice loss on NAO- events only. As stated in l.144, the NAO only explains one third of the 1.5-meter temperature variability over Northern Europe, and as you focus on the negative phase only that represents even less than that (only ~70 winters out of 500). Other weather patterns impact the climate of Northern Europe, especially the blocking systems, whose intensity/frequency can be impacted by sea-ice loss (e.g. Francis and Vavrus 2015). Does the conclusion that the thermodynamic warming due to sea ice loss offsets the dynamical response apply in other winters than the ones that are dominated by NAO-? That's an important question that has to be discussed in the paper. I suggest that you at least show and discuss the mean response to sea ice loss (for all 501 winters) so we know whether the absence of cooling over Northern Europe is common to all winters (Figures could be provided in SI). In order to give a more complete picture, you could also decompose the winter circulation in weather regime patterns (e.g. Hertzog and Jacobbeil 2014), or self-organized maps (Reusch et al. 2007), then perform the same thermodynamical versus dynamical decomposition that you do for NAO-. It would be very interesting to verify that the missing cooling is a robust feature of decreased sea ice experiment, not only for NAO-, but also for blockings for example. I understand such an analysis can be considered to be beyond the scope of the study, but at least that should be discussed in the paper.

Hertzog, E. and Jacobbeil, J. (2014), Variability of weather regimes in the North Atlantic-European area: past and future. *Atmos. Sci. Lett.*, 15: 314–320. doi:10.1002/asl2.505

Reusch, D. B., R. B. Alley, and B. C. Hewitson (2007), North Atlantic climate variability from a self-organizing map perspective, *J. Geophys. Res.*, 112, D02104, doi:10.1029/2006JD007460.

- How are the ensemble mean and standard deviation of the NAO events defined (Methods section) ? You find as many NAO- events in LI and HI (l. 362) so I guess you use the ensemble mean and standard deviation from each experiment (otherwise we'd find less NAO- winters in LI). Therefore, do the changes that you show represent a change in extreme NAO- events (following your -1std dev definition), or a change in the mean state of the distribution ? Extreme NAO- events in LI are expected to be more intense if they are defined from a mean state that is more NAO- like. That is a different conclusion than saying that extreme NAO events are more intense in LI, which is the message I get from your study. In other words, does the mean of the distribution change, or its shape ? That needs to be clarified.

- I think the polar cap height (PCH) response in Fig. 2c has to be interpreted with caution. The "late-winter" NAO- events are defined over the January-February period (to me JF is rather mid-winter by the way), so by construction we expect to find a warm PCH response in the stratosphere over this period, given the relationship between a weak polar vortex and NAO- events at the surface. It does not mean that the overall response to the sea ice anomalies (i.e. not only for JF NAO- winters) is maximum over January-February, it could as well be stronger later in winter, in February-March or March alone for example (several studies have shown such a late winter response to sea ice loss, e.g. Peings and Magnusdottir 2014, Sun et al. 2015). It is thus possible that the maximum response to sea ice occurs later in winter and that it is accompanied with a significant cooling over Northern Europe (that is not found in your composites maybe due to a weaker response of the PCH in JF). That should be discussed to make sure you are not missing the cooling effect of sea ice by focusing on JF. I'd like to see these two questions answered : Is January-February the "optimum" time-window to detect the potential cooling response to sea ice loss ? What is the dependence of the results to the choice of this time-window (what about February-March for example) ?

- As mentioned in Methods, you prescribe different SST background states during the course of the simulations (300 years with PDO cycle, 202 with AMO cycle). What is the dependence of the results to the PDO/AMO prescription ? You mention in introduction that the response to sea ice is dependent on the background SST, so that should be discussed in the paper.

- I feel the conclusion can be improved. The statement that "European temperatures are not governed by the NAO", in l. 298, is misleading. To me, on the contrary your study clearly shows that the NAO controls the European temperatures by advecting air from the subpolar regions. If not, Northern Europe would exhibit warm anomalies in response to sea ice loss, as do the northern latitudes of Siberia and North America (Fig. 4c and 6b). The dynamical effect of the NAO- response is to bring polar air masses over Europe. Although anomalously warm due to the loss of sea ice, these air masses mitigate the warm response expected from the thermodynamical effect only, leading to insignificant changes of the surface temperature. In this sense the title of your study could almost be inverted ("The missing Northern European winter warming response to Arctic sea ice loss") as it is clear on Fig. 4c and 6b that the primary effect of sea ice loss is to warm the adjacent continents in the mid-high latitudes. But I understand that the present title is more in phase with the current debate on the Arctic-mid-latitude linkages.

Anyway, I think the conclusion should be remodeled to clarify that the NAO- response controls the European temperature, but that the dynamical anomaly does not necessarily lead to a cooling since the mid-latitude temperature response depends on the nature of the advected air masses. In LI, these masses are not cold enough (due to the local effect of sea ice loss) to significantly cool Northern Europe hence the cooling effect of the NAO- is absent. That does not mean that the NAO- is not a skillful predictor of the surface temperature (l. 295), your study rather suggests that the NAO index should be coupled with a predictor of advected air temperature to forecast surface temperature anomalies in the mid-latitudes.

- In link with my previous comment, in the real world, other surface anomalies than sea ice loss that are not accounted in idealized experiments such as the ones from this study (for example

snow cover). Such anomalies can supplement with the effect of sea ice anomalies and modify the temperature of air masses that are advected by the dynamical response to sea ice. An anomalous snow extent can lead to colder air masses and to more pronounced cooling anomalies over Northern Europe, especially over the present period when the decrease in snow extent is not as pronounced than in late 21st century projections. I think the fact that your experiments neglect the possible impact of other sources of climate variability on the NAO-advected air masses has to be mentioned.

Minor comments

- l. 84-88 : as mentioned in a previous comment, does the changes in NAO describe a change in mean state of the mid-latitude flow, or an increase in extreme NAO events ? That should be clarified here.

- l.105 and 108, Fig. 1c is Fig. 2c.

- Could you indicate the spatial correlation between the contours and shading in Fig. 3 ? You could also give it for other months so we get an idea of the phasing between the anomalous and the background wave during the other months (l. 131).

- Fig. 5d suggests that stronger NAO- events in LI (easterly flow lower than -5 m/s) induce a cooling, which seems contradictory with the fact that cold extreme decrease in LI (Fig. 5b). Could you clarify this section ?

- You find that the dynamical response to sea ice loss is smaller in the C21/C20 set of experiments (l. 253). Is that due to a smaller response of PCH in the stratosphere ? It would be interesting to show (in SI?) and discuss it. You could also provide the spatial correlation between anomalous vs background wave-1 in C21 vs C20 to determine if a smaller stratospheric response is due to less constructive or even destructive interferences (maybe in a table that would also indicate month-by-month correlations for LI vs HI, see my previous comment).

- In Fig. 6e, the wind-temperature relationship is different in C21 and C20 (unlike LI vs HI, the slope is different). In C21 compared to C20, a similar change in zonal wind has less impact on the temperature, probably due to the fact that the mid-latitude vs pole gradient of surface temperature is largely decreased in C21. This linear relationship is computed from all days, not only NAO- days, right ? What is DT/DU in C21 (I understand that the one you give is for C20), and how does it compare to the -0.08°C cooling you find in C21 vs C20 for NAO- winters ? Is the decreased relationship between temperature and wind stronger during NAO- events, or is it similar to the overall change simulated in C21 ? Again, the discussion of the C21/C20 experiments has to be clarified (l. 233-263).

- The last sentence of the abstract is too strong ("daily and multi-day cold extremes decrease and their changes are not governed by the NAO"). Again, to me the fact that the "expected" cooling response of the NAO is missing does not mean that the NAO does not govern changes in temperatures. Fig. 5d shows a clear dependence of the temperature response to the intensity of zonal wind (hence the NAO).

Reviewer #2 (Remarks to the Author):

Review of the manuscript 'The missing Northern European winter cooling response to Arctic sea ice loss'
by Screen

The author uses large-ensemble atmosphere-only simulations with prescribed sea surface

temperatures and sea-ice concentrations to examine how seasonal-scale NAO- events are affected by Arctic sea ice loss. He compares simulations with low and high Arctic sea-ice. Despite an intensification of NAO- events, reflected by more prevalent easterly flow, sea ice loss does not lead to Northern European winter cooling. The dynamical cooling from the changed NAO is missing because it is offset by thermodynamical warming.

Major Comments:

- whether Arctic sea-ice loss has a substantial effect on the NAO might be strongly model dependent and may also depend on the exact boundary conditions - it seems to be a back and forth between different studies (see e.g. the literature mentioned in lines 39-42).

- the author focuses on the intensity changes of NAO- events and their effects on temperature. There seems to be no substantial frequency changes in NAO- events in the LI/HI ensemble simulations - in both are 71 NAO- events (see Method section line 362). In contrast to these more moderate changes in sea ice the author compares two sets of simulations with stronger sea-ice changes (C21 and C20). In contrast to the LI/HI simulations there are changes in the frequency of NAO- events between the C21 and C20 ensembles. The C21 ensemble has more NAO- events than the C20 ensemble - 47 compared to 43 (see Method section line 362). Therefore, it seems to me that Arctic sea-ice reductions can potentially cause colder winters in Europe. This frequency change in NAO- events between the C21 and C20 and its effect is not discussed in the paper.

- computation of the NAO index: I would prefer to compute the NAO index using EOF analysis because this would account for possible differences in the location of the centers of actions in the different ensembles.

- regarding the NAO computation in the paper using SLP differences. I would prefer to normalize/standardize the two SLP regions first before computing the difference. I would expect that the Icelandic Low has a higher variability than the Azores High in the simulations. Otherwise the NAO index is dominated by the variability of the northern box (Icelandic Low). see also previous comment

Reviewer #3 (Remarks to the Author):

The present manuscript by James Screen discusses if the loss of Arctic sea ice leads to increased winter cooling in the European region related to negative NAO events. While an intensification of negative NAO events with intensified easterly wind anomalies is found, the study finds an compensation by additional warming leading to almost no cooling effect over Europe. This is in contradiction to many observational results. The study adds to one of the most discussed recent topics in climate sciences and is therefore highly relevant for publishing.

As I will explain in my detailed comments, I have some doubts about the accuracy of either the description of methods or the implementation of methods. Furthermore, I feel that the conclusions are driven by the somewhat prejudiced aim to prove that no cooling effect exists. If the methods are robust, I would vote for minor revisions. At the moment I have doubts about the robustness and therefore vote for major revisions.

General comment on the determination of NAO events: If I understand right, NAO events have been calculated from a two month average. This is a quite rough approach to detect extremes that may last only some weeks but unlikely two month. You may average over additional warm periods. Additionally, you underestimate events that do not (fully) extent over that period. If the typical date of such extremes changes due to the influence of sea ice changes, your approach would also lead to questionable conclusions. Furthermore, the implementation of two domain averages instead of more common statistical methods like EOF analysis is unexpectedly simple. I am not fully convinced by the simplicity of your statistics and your rather rough approach.

NAO calculation details: It is not clear from lines 360-361, if the ensemble mean and ensemble standard deviation is calculated for LI (C21) and HI (C20) separately or in common. This is really important, since mean and standard deviation likely differ between both ensembles. Thus, using different offset and scaling for both ensembles potentially contaminates parts of the difference seen in Table 1 and Figure 2. You need to implement the same climatology for both ensembles or you have to account for their differences.

SST in ensembles: SST is derived from climatology, while in regions close to the ice edge (maybe the most sensitive area) additional anomalies are applied. At the grid points where both climatology and additional anomalies meet, is there any smoothing? There could be unrealistic or inconsistent jumps that lead to unrealistic or disturbed forcing. Additionally, you added years with additional multi-decadal variations related to AMO and PDO – which is great. But, I do not see the reason behind adding the PDO in 150 members and the AMO in 101 members. This should be explained. Are there members where AMO and PDO vary simultaneously?

Lines 79-82: I don't see any difference that is higher in LI minus HI compared to C21 minus C20. C21 seems to have almost no sea ice at all in early winter (according to the red line). Therefore, the difference between C21 and C20 is actually very large. The only thing I see is that HI has some more ice than C20. This comes not unexpected, since many CMIP5 models show too less sea ice compared to observations in early years.

Line 132: It is not essentially necessary that the sea ice directly forces the waves in November. Tropospheric conditions force these waves, but these conditions can be the result of accumulated forcing not only from November (and not only generated by sea ice).

Lines 173-176: Again I ask myself which climatology has been taken. Is it for each ensemble separately? Is it a common climatology? If the climatology differs between both ensembles (which is very likely), you have to account for that!

Lines 186-189: This is inaccurately written. From figure 5c one clearly sees that the linear relationship is different between LI and HI. Which dT/dU has been taken to calculate this change (with an unrealistic precision of two decimals without any estimate of error)? Since both gradients differ, the temperature difference depends on the wind bin. This should be accounted for.

Actually, this is missing in the whole manuscript: error estimation. You provide two decimals most of the time but no estimate of the error interval. The only discussion of errors / discussion of significance appear in Table 1 and 2, last column.

Figure 5c, line 208: Something must be wrong here. For a given wind bin, HI (red) shows higher temperatures than LI (blue) according to your caption but in contrast to line 208. The red line in

Figure 5d suggests that the caption for figure 5c is wrong, since ΔT must be positive and you actually state that ΔT is positive in Table 1. Please double check your data and figures!

Lines 213-216: I would object to parts of that conclusion: Especially the negative zonal wind anomalies get even cooler or stay as cool as they are. Actually, these characterize the extremes. Therefore, one has to state that the extreme negative NAO events stay as cold or become even colder. Especially the dynamical contribution is negative and is only partly cancelled out by the thermodynamical contribution. The conclusion that no cooling effect exists holds for the extreme negative NAO events of future projections (referring to figure 6f), although the dynamical effect is still negative. This leads me to the question, if the temperature variability is increased? In fact, this would imply that cold extremes might be not as cold as today measured by absolute temperature, but very similar. If, simultaneously, the general variability is increased the extremes become more extreme measured by the actual average temperature, this would be an increase of extremes anyway. In terms of adaptation: We have to adapt to significantly warmer climate, but winter cold extremes stay as cold as in present climate. Thus, the conclusion significantly depends on your definition of extremes. Although it is right to claim that cold extremes do not become colder according to your data, the former aspects need some discussion!

Line 225-231: This whole paragraph is just a citation of ref. 64? I see only minor contributions from the present study to these results. Maybe this should be moved to conclusions or should be removed.

Line 248: Fig. 3c does not exist.

Reviewer #1 (Reviewer comments in blue; Responses in black)

This paper is a modeling study that explores the atmospheric response to Arctic sea ice loss using a large ensemble of sensitivity numerical experiments. The originality of the study lies in investigating causes for the apparent inconsistency between the dynamical response to sea ice loss (negative NAO-like pattern) and the near-surface temperature response that exhibits little or no winter cooling over Northern Europe. The author shows evidence that the thermodynamical effect of sea ice loss, i.e. a surface warming at the pole and in adjacent regions, offsets the effect of the dynamical response due to advection of relatively warmer air when sea ice is decreased.

The study provides a interesting contribution to the field, that is going to help interpreting the temperature response to Arctic sea-ice loss in other similar numerical studies. Anticipating the effect of Arctic Amplification on the mid-latitude climate is a great challenge for climate scientists, and this study highlights an important aspect, that is that a loss of Arctic sea ice does not necessarily lead to cooler winter over Europe as expected from dynamical considerations. The paper is well-written, but I think some aspects could be clarified, especially concerning the methodology and interpretation of the results. In particular, I'd like to see a discussion of the overall impact of sea ice loss in other winters than NAO- ones, and also clarify whether the conclusion of this study applies to other weather patterns than NAO-. The discussion and main conclusions can be improved too.

Please see hereafter a list of comments. I am confident that the paper will be suitable for publication after improving these points, I therefore recommend a major revision.

I thank the Reviewer for their time, thoughtful assessment and excellent suggestions. Responding to this feedback has significantly improved the manuscript.

Major comments

- My first major comment concerns the fact that the study focuses on the impact of sea ice loss on NAO- events only. As stated in I.144, the NAO only explains one third of the 1.5-meter temperature variability over Northern Europe, and as you focus on the negative phase only that represents even less than that (only ~70 winters out of 500). Other weather patterns impact the climate of Northern Europe, especially the blocking systems, whom intensity/frequency can be impacted by sea-ice loss (e.g. Francis and Vavrus 2015). Does the conclusion that the thermodynamic warming due to sea ice loss offsets the dynamical response applies in other winters than the ones that are dominated by NAO- ? That's an important question that has to be discussed in the paper. I suggest that you at least show and discuss the mean response to sea ice loss (for all 501 winters) so we know whether the absence of cooling over Northern Europe is common to all winters (Figures could be provided in SI). In order to give a more complete picture, you could also decompose the winter circulation in weather regime patterns (e.g. Hertig and Jacobeit 2014), or self-organized maps (Reusch et al. 2007), then perform the same thermodynamical versus dynamical decomposition that you do for NAO-. It

would be very interesting to verify that the missing cooling is a robust feature of decreased sea ice experiment, not only for NAO-, but also for blockings for example. I understand such an analysis can be considered to be beyond the scope of the study, but at least that should be discussed in the paper.

I recognise that the NAO is only one weather pattern that affects Northern Europe. I chose to focus specifically on the NAO- for a couple of key reasons.

Firstly, in these model simulations the overall midwinter circulation response resembles the NAO- (see new Supplementary Fig. 1) and hence, I wanted to look more closely at this specific atmospheric phenomena. I have looked at other phenomena, such as blocking and NAO+, but haven't found robust evidence for a simulated change in response to Arctic sea ice loss. For example, Ayarzagüena and Screen (2016) looked at cold-air outbreaks (often connected to blocking) and found little evidence of dynamical changes in blocking. Other unpublished analyses confirm this general conclusion. We are also in the process of undertaking a weather regime analysis using self-organizing maps (SOMs). Preliminary analyses of these results suggest that over Europe the only weather regime to show a noteworthy increase in its frequency of occurrence is a pattern that resembles the NAO-. I agree with the Reviewer that such a regime analysis, whilst potentially insightful, goes beyond the scope of the present manuscript (which is intended to be a fairly short, focused contribution).

Secondly, considering not just my simulations but also the wider literature, the one dynamical change that appears common (if not completely robust) in response to sea ice loss is a shift towards the NAO- (or AO-). My feeling is that hypothesized changes in blocking (e.g., Francis and Vavrus, 2015) are not supported by the majority of modeling studies. The prevalence of a NAO- response in modeling and observational studies has led to the NAO- becoming an analogue for the predicted response to sea ice loss (in a way that other weather patterns have not). This further motivated focusing on the NAO-.

Reflecting on this comment I have made the following changes:

- **I have added a plot of the overall response (Supplementary Fig. 1) and noted in the text that the NAO- response and lack of Northern European cooling are also visible in this.**
- **I have added a sentence at the start of the Results section that further emphasizes the motivation for focusing on the NAO-.**
- **I have added a sentence in the Discussion that states that further work is required to see if the conclusions for the NAO- also apply to other weather patterns.**

- How are the ensemble mean and standard deviation of the NAO events defined (Methods section) ? You find as many NAO- events in LI and HI (l. 362) so I guess you use the ensemble mean and standard deviation from each experiment (otherwise we'd find less NAO- winters in LI). Therefore, do the changes that you show represent a change in extreme NAO- events (following your -1std dev definition), or a change in the mean state of the distribution ? Extreme NAO- events in LI are expected to be more intense if they are defined from a mean state that is more NAO- like. That is a different conclusion than saying that extreme NAO events

are more intense in LI, which is the message I get from your study. In other words, does the mean of the distribution changes, or its shape ? That needs to be clarified.

I apologise, this section of the Methods was a bit ambiguous. I have added a sentence that explicitly states that the ensemble mean and standard deviation were calculated separately for each experiment. I have also undertaken some sensitivity analyses (see Supplementary Fig. 8) that confirm that the results are not sensitive to this choice, comparing to an alternative approach of normalising relative to HI. In the latter case there are *more* NAO- winters identified in LI because the standard deviation is increased relative to HI.

In answer to the Reviewer's repeated question regarding whether it is the mean state or the shape of the NAO distribution that changes, I can confirm that it is predominantly the latter. There is no significant change in the mean but an increase in variability, which is entirely consistent with an intensification of NAO- events. A sentence has been added to the text on this.

- I think the polar cap height (PCH) response in Fig. 2c has to be interpreted with caution. The "late-winter" NAO- events are defined over the January-February period (to me JF is rather mid-winter by the way), so by construction we expect to find a warm PCH response in the stratosphere over this period, given the relationship between a weak polar vortex and NAO- events at the surface. It does not mean that the overall response to the sea ice anomalies (i.e. not only for JF NAO- winters) is maximum over January-February, it could as well be stronger later in winter, in February-March or March alone for example (several studies have shown such a late winter response to sea ice loss, e.g. Peings and Magnusdottir 2014, Sun et al. 2015). It is thus possible that the maximum response to sea ice occurs later in winter and that it is accompanied with a significant cooling over Northern Europe (that is not found in your composites maybe due to a weaker response of the PCH in JF). That should be discussed to make sure you are not missing the cooling effect of sea ice by focusing on JF. I'd like to see these two questions answered : Is January-February the "optimum" time-window to detect the potential cooling response to sea ice loss ? What is the dependence of the results to the choice of this time-window (what about February-March for example) ?

I totally agree that Fig. 2c can only be interpreted in the context of midwinter NAO- events (note: I've followed the suggestion to rename late winter as midwinter). I've made sure that each instance of "NAO- event" is preceded by "midwinter" in this paragraph, to be explicitly clear this is what is shown. In practice however, the overall PCH response (i.e., in all winters) is very similar to that shown in Fig. 2c.

Although it wasn't stated explicitly before (it is now), I decided to analyse Jan-Feb precisely because this is the optimum time window to detect the potential cooling response. It is in these months that the intensification of NAO- events is found in these simulations, as can be seen in Supplementary Fig. 3. I'm aware, as the Reviewer points out, that other studies have found a maximum response in Feb-Mar. This suggests the timing of the NAO response is either model dependent or sensitive to the precise details of the sea ice forcing. I've added a comment on this in the revised text.

- As mentioned in Methods, you prescribe different SST background states during the course of the simulations (300 years with PDO cycle, 202 with AMO cycle). What is the dependence of the results to the PDO/AMO prescription ? You mention in introduction that the response to sea ice is dependent on the background SST, so that should be discussed in the paper.

It turns out that the results are not sensitive to the background state in this case, as can be seen in Supplementary Figs. 4 and 5. The NAO- response and absence of Northern European cooling are features of all four sub-samples (sampled based on SST state). This is now discussed. Other features of the response (not relevant to this paper) do however appear sensitive to the background state, for example, the circulation response over North America (the subject of a separate paper currently in review elsewhere).

- I feel the conclusion can be improved. The statement that “European temperatures are not governed by the NAO”, in l. 298, is misleading. To me, on the contrary your study clearly shows that the NAO controls the European temperatures by advecting air from the subpolar regions. If not, Northern Europe would exhibit warm anomalies in response to sea ice loss, as do the northern latitudes of Siberia and North America (Fig. 4c and 6b). The dynamical effect of the NAO- response is to bring polar air masses over Europe. Although anomalously warm due to the loss of sea ice, these air masses mitigate the warm response expected from the thermodynamical effect only, leading to insignificant changes of the surface temperature. In this sense the title of your study could almost be inverted (“The missing Northern European winter warming response to Arctic sea ice loss”) as it is clear on Fig. 4c and 6b that the primary effect of sea ice loss is to warm the adjacent continents in the mid-high latitudes. But I understand that the present title is more in phase with the current debate on the Arctic-mid-latitude linkages. Anyway, I think the conclusion should be remodeled to clarify that the NAO- response controls the European temperature, but that the dynamical anomaly does not necessarily lead to a cooling since the mid-latitude temperature response depends on the nature of the advected air masses. In LI, these masses are not cold enough (due to the local effect of sea ice loss) to significantly cool Northern Europe hence the cooling effect of the NAO- is absent. That does not mean that the NAO- is not a skillful predictor of the surface temperature (l. 295), your study rather suggests that the NAO index should be coupled with a predictor of advected air temperature to forecast surface temperature anomalies in the mid-latitudes.

On reflection I completely agree, my choice of wording could have been better. I have rephrased the relevant sections of the conclusions and abstract.

- In link with my previous comment, in the real world, other surface anomalies than sea ice loss that are not accounted in idealized experiments such as the ones from this study (for example snow cover). Such anomalies can supplement with the effect of sea ice anomalies and modify the temperature of air masses that are advected by the dynamical response to sea ice. An anomalous snow extent can lead to colder air masses and to more pronounced cooling anomalies over Northern Europe, especially over the present period when the decrease in snow extent is not as pronounced than in late 21st century projections. I think the fact that your experiments neglect the possible impact of other sources of climate variability on

the NAO-advected air masses has to be mentioned.

This is a fair point. I've mentioned this caveat in the Discussion.

Minor comments

- l. 84-88 : as mentioned in a previous comment, does the changes in NAO describe a change in mean state of the mid-latitude flow, or an increase in extreme NAO events ? That should be clarified here.

Please refer to response above.

- l.105 and 108, Fig. 1c is Fig. 2c.

Thank you. This has been corrected.

- Could you indicate the spatial correlation between the contours and shading in Fig. 3 ? You could also give it for other months so we get an idea of the phasing between the anomalous and the background wave during the other months (l. 131).

Thank you for this excellent suggestion. I've added a second panel to Fig. 3 which shows the spatial correlation between the forced and climatological waves as a function of month.

- Fig. 5d suggests that stronger NAO- events in LI (easterly flow lower than -5 m/s) induce a cooling, which seems contradictory with the fact that cold extreme decrease in LI (Fig. 5b). Could you clarify this section ?

I agree this requires clarification. This apparent discrepancy arises because the zonal wind is only one factor that influences temperature (the meridional wind, cloud cover, upstream conditions and so on, all play a role). This means that there is not a one-to-one relationship between the wind and temperature PDFs (i.e., coldest days aren't necessary the most easterly), despite the strong wind-temperature relationship. The coldest days are not all associated with strong easterlies. To give some specific numbers, days more than 10C colder than average in HI have zonal wind values ranging from -5.8 m/s to 5.2 m/s (mean of -1.5 m/s). This tells us that some of the coldest days occur in wind categories that are warming; hence the reduction in cold extremes is not contradictory to Fig. 5d. I've added a couple of sentences explaining this.

- You find that the dynamical response to sea ice loss is smaller in the C21/C20 set of experiments (l. 253). Is that due to a smaller response of PCH in the stratosphere ? It would be interesting to show (in SI?) and discuss it. You could also provide the spatial correlation between anomalous vs background wave-1 in C21 vs C20 to determine if a smaller stratospheric response is due to less constructive or even destructive interferences (maybe in a table that would also indicate month-by-month correlations for LI vs HI, see my previous comment).

During the revision I noticed a small bug in my code that meant that a subset of NAO- cases were being incorrectly chosen. This only affected Figures 5 and 6.

Having corrected this I now find that the dynamical response is comparable in both sets of simulations. Therefore, this comment has become redundant. However, I have added a note in the text that the dynamical response does not scale linearly with the loss of sea ice, with citations to a few studies that have drawn similar conclusions. It is beyond the scope of the present manuscript to identify the cause of this nonlinearity.

- In Fig. 6e, the wind-temperature relationship is different in C21 and C20 (unlike LI vs HI, the slope is different). In C21 compared to C20, a similar change in zonal wind has less impact on the temperature, probably due to the fact that the mid-latitude vs pole gradient of surface temperature is largely decreased in C21. This linear relationship is computed from all days, not only NAO- days, right ? What is DT/du in C21 (I understand that the one you give is for C20), and how does it compare to the -0.08°C cooling you find in C21 vs C20 for NAO- winters ? Is the decreased relationship between temperature and wind stronger during NAO- events, or is it similar to the overall change simulated in C21 ? Again, the discussion of the C21/C20 experiments has to be clarified (l. 233-263).

The Reviewer raises a good point about the differing slopes of the temperature-wind relationship in C21 and C20. I agree this is consistent with reduced horizontal temperature gradients in C21 versus C20. I've added a comment on this in the revised text.

The bin averages and linear relationships in Fig. 5c and Fig. 6e are for all days in NAO- midwinters, not all days in all midwinters. This has been clarified in the figure caption and text. The du value is the difference between C21 and C20 (or HI and LI). I then use du to estimate the expected temperature response (based on the linear temperature-wind relationship). This has been clarified in the caption to Fig. 5.

- The last sentence of the abstract is too strong ("daily and multi-day cold extremes decrease and their changes are not governed by the NAO"). Again, to me the fact that the "expected" cooling response of the NAO is missing does not mean that the NAO does not govern changes in temperatures. Fig. 5d shows a clear dependence of the temperature response to the intensity of zonal wind (hence the NAO).

As noted above, my use of the wording "not governed" was poorly conceived. I have rephrased the last two sentences of the abstract.

Reviewer #2 (Reviewer comments in blue; Responses in black)

The author uses large-ensemble atmosphere-only simulations with prescribed sea surface temperatures and sea-ice concentrations to examine how seasonal-scale NAO- events are affected by Arctic sea ice loss. He compares simulations with low and high Arctic sea-ice. Despite an intensification of NAO- events, reflected by more prevalent easterly flow, sea ice loss does not lead to Northern European winter cooling. The dynamical cooling from the changed NAO is missing because it is offset by thermodynamical warming.

I thank the Reviewer for their time and useful suggestions.

Major Comments:

- whether Arctic sea-ice loss has a substantial effect on the NAO might be strongly model dependent and may also depend on the exact boundary conditions - it seems to be a back and forth between different studies (see e.g. the literature mentioned in lines 39-42).

I agree that the results may be model dependent and have added a comment on this. However, as noted in the text, a NAO- response is common (if not fully robust) in many model (and observational) papers.

- the author focuses on the intensity changes of NAO- events and their effects on temperature. There seems to be no substantial frequency changes in NAO- events in the LI/HI ensemble simulations - in both are 71 NAO- events (see Method section line 362). In contrast to these more moderate changes in sea ice the author compares two sets of simulations with stronger sea-ice changes (C21 and C20). In contrast to the LI/HI simulations there are changes in the frequency of NAO- events between the C21 and C20 ensembles. The C21 ensemble has more NAO- events than the C20 ensemble - 47 compared to 43 (see Method section line 362). Therefore, it seems to me that Arctic sea-ice reductions can potentially cause colder winters in Europe. This frequency change in NAO- events between the C21 and C20 and its effect is not discussed in the paper.

The number of NAO- events is a product of the method of defining these events relative to the mean and standard deviation of each ensemble (i.e., a fixed frequency threshold). Give this approach, the equal number of events in HI and LI is unexpected. More surprising is the unequal number of events in C21 and C20, which on closer inspection, relates to differences in higher-order moments (more negative skewness and more negative kurtosis). I've added a sentence explaining this to the Methods.

The question of whether it's more appropriate to use a fixed frequency threshold (as was done) or a fixed NAO threshold is a philosophical one. Since the NAO is effectively a measure of anomalous circulation, it seems justifiable to define its extremes relative to the climatology of each ensemble. An alternative approach however, would be to normalise relative to a fixed NAO mean and standard deviation (i.e., from HI for both HI and LI, and from C20 for both C20 and C21). Doing this yields more events in LI than HI (85 vs. 71), and in C21 than C20 (67 vs. 42) because the mean state becoming more like the NAO-. So relative to a common reference, both sets of simulations show an increase in NAO- events due to sea ice loss. Does this frequency change lead to cooling over Northern Europe? One way to assess this is to look at the climatological temperature response for all midwinters (not just NAO- ones). As can be seen in Supplementary Fig. 1 (for LI vs. HI) and Supplementary Fig. 6 (for C21 vs. C20), the climatological response shows no significant cooling over Northern Europe, despite a climatological circulation change that resembles the NAO- (in this case a reflection of both the increase in frequency and intensity of NAO- events). In summary, the differences in frequency

(depending on how extremes are defined) do not effect the overall conclusion that NAO-driven cooling is offset by thermodynamical warming.

- computation of the NAO index: I would prefer to compute the NAO index using EOF analysis because this would account for possible differences in the location of the centers of actions in the different ensembles.
- regarding the NAO computation in the paper using SLP differences. I would prefer to normalize/standardize the two SLP regions first before computing the difference. I would expect that the Icelandic Low has a higher variability than the Azores High in the simulations. Otherwise the NAO index is dominated by the variability of the northern box (Icelandic Low). see also previous comment.

I have experimented with alternative NAO definitions, including EOF-based approaches and the results are insensitive to this (see Supplementary Fig. 7). A sentence has been added to the Methods section on this. For the main text I have chosen to retain the original SLP difference index, for reasons of simplicity (it is easy to communicate, especially to non-experts) and because I feel a pressure difference retains more physical meaning (e.g., it can be directly related to the wind field) compared to EOFs which are statistical constructs.

Reviewer #3 (Reviewer comments in blue; Responses in black)

The present manuscript by James Screen discusses if the loss of Arctic sea ice leads to increased winter cooling in the European region related to negative NAO events. While an intensification of negative NAO events with intensified easterly wind anomalies is found, the study finds an compensation by additional warming leading to almost no cooling effect over Europe. This is in contradiction to many observational results. The study adds to one of the most discussed recent topics in climate sciences and is therefore highly relevant for publishing.

I thank the Reviewer for their time and useful feedback.

As I will explain in my detailed comments, I have some doubts about the accuracy of either the description of methods or the implementation of methods. Furthermore, I feel that the conclusions are driven by the somewhat prejudiced aim to proof that no cooling effect exists. If the methods are robust, I would vote for minor revisions. At the moment I have doubts about the robustness and therefore vote for major revisions.

With regards to the methods, I believe it is the former (unclear description of methods) as opposed to the latter (implementation of methods). As detailed further below, the relevant aspects of the methods have been clarified. The accusation of a “prejudiced aim” is unjustified and something I strongly contest. Furthermore contrary to Reviewer’s implication, the manuscript does not prove that no cooling

effect exists. On the contrary, it demonstrates the NAO- shift induces a cooling effect but that this is offset by a thermodynamical warming effect. As Reviewer 1 pointed out, and I fully acknowledge, my previous usage of the wording “temperature response is not governed by the NAO” was poorly chosen. The revised manuscript clarifies that the NAO- does play a key role (a cooling effect) but that this is offset by a warming effect.

General comment on the determination of NAO events: If I understand right, NAO events have been calculated from a two month average. This is a quite rough approach to detect extremes that may last only some weeks but unlikely two month. You may average over additional warm periods. Additionally, you underestimate events that do not (fully) extent over that period. If the typical date of such extremes changes due to the influence of sea ice changes, your approach would also lead to questionable conclusions. Furthermore, the implementation of two domain averages instead of more common statistical methods like EOF analysis is unexpectedly simple. I am not fully convinced by the simplicity of your statistics and your rather rough approach.

This is correct, the NAO events are defined based on two-month averages. The text is explicit in this regard, but I’ve added a further statement of this fact (and justification) near the start of the Results. Of course, I recognise that within an NAO- midwinter there will be shorter periods (days, or weeks) that are strongly NAO- and those that are not. This is reflected in the histograms of daily zonal wind (Fig. 5a; i.e., not all days within a NAO- winter are characterised by easterly flow). And yes, these events include shorter warm periods (see Fig. 5b), but that doesn't invalidate the approach taken. An “extreme” is a relative construct. Here I am concerned with seasonal (2-month) mean NAO- extremes.

An EOF-based definition of the NAO yields near-identical results (Supplementary Fig. 7).

NAO calculation details: It is not clear from lines 360-361, if the ensemble mean and ensemble standard deviation is calculated for LI (C21) and HI (C20) separately or in common. This is really important, since mean and standard deviation likely differ between both ensembles. Thus, using different offset and scaling for both ensembles potentially contaminates parts of the difference seen in Table 1 and Figure 2. You need to implement the same climatology for both ensembles or you have to account for their differences.

I apologise, this section of the Methods was a bit ambiguous. I have added a sentence that explicitly states that the ensemble mean and standard deviation were calculated separately for each experiment. I have also undertaken some sensitivity analyses (see Supplementary Fig. 8) that confirm that the results are not sensitive to this choice, comparing to an alternative approach of normalising relative to HI. For the record, there is no significant change in the mean NAO index but an increase its variability, which is entirely consistent with an intensification of NAO- events. A sentence has been added to the text on this.

SST in ensembles: SST is derived from climatology, while in regions close to the ice edge (maybe the most sensitive area) additional anomalies are applied. At the grid

points where both climatology and additional anomalies meet, is there any smoothing? There could be unrealistic or inconsistent jumps that lead to unrealistic or disturbed forcing. Additionally, you added years with additional multi-decadal variations related to AMO and PDO – which is great. But, I do not see the reason behind adding the PDO in 150 members and the AMO in 101 members. This should be explained. Are there members where AMO and PDO vary simultaneously?

No smoothing has been applied. Unrealistic SST gradients are an unavoidable consequence of this type of AGCM experiment. However, through many years of experience of performing such experiments, these gradients appear to be of little concern. There is no evidence in the literature, or from my own extensive experience, that these gradients have any noticeable effect of the large-scale circulation response to sea ice loss (one could imagine they might have some small-scale convective effects close to the ice edge, but the horizontal resolution of this model is probably too coarse to capture such things).

The reason for fewer AMO members than PDO members is purely due to available computing resources. There are not members where the AMO and PDO are explicitly varied simultaneously; however, by the method to calculate the SST anomalies does allow for correlation between the AMO and PDO. So, for example, the AMO members do have SST anomalies in the Pacific (although relatively small compared to the Atlantic) that reflect the observed correlation between the AMO and PDO. It turns out that the results are not sensitive to the background state in this case, as can be seen in Supplementary Figs. 4 and 5. The NAO- response and absence of Northern European cooling are features of all four sub-samples (sampled based on SST state). This is now discussed.

Lines 79-82: I don't see any difference that is higher in LI minus HI compared to C21 minus C20. C21 seems to have almost no sea ice at all in early winter (according to the red line). Therefore, the difference between C21 and C20 is actually very large. The only thing I see is that HI has some more ice than C20. This comes not unexpected, since many CMIP5 models show too less sea ice compared to observations in early years.

The differences in sea-ice loss are clearer when they are plotted explicitly (see Supplementary Fig. 2).

Line 132: It is not essentially necessary that the sea ice directly forces the waves in November. Tropospheric conditions force these waves, but these conditions can be the result of accumulated forcing not only from November (and not only generated by sea ice).

I take your point: even though the response is largest in November this doesn't necessarily mean it is driven by sea-ice loss in November. I've reworded this sentence.

Lines 173-176: Again I ask myself which climatology has been taken. Is it for each ensemble separately? Is it a common climatology? If the climatology differs between both ensembles (which is very likely), you have to account for that!

The question of whether it is more appropriate to use a common climatology or a varying climatology (and how to define an “extreme”; relative to what?) is a philosophical one. Here the temperature anomalies were defined relative to a common climatology (from HI). This was stated in the figure caption and is now reiterated in the text. Remember, there is very little change in Northern European mean temperature, so in practice it makes very little difference if the anomalies are calculated relative to the mean of each ensemble separately (see figure below). The reduction in cold extremes stems partly from mean warming but also from reduced variability, as discussed in the text, and is clearly depicted in histograms of anomalies relative to a common climatology (panel a; same as Fig. 5b) or differing climatology (panel b).

Lines 186-189: This is inaccurately written. From figure 5c one clearly sees that the linear relationship is different between LI and HI. Which dT/dU has been taken to calculate this change (with an unrealistic precision of two decimals without any estimate of error)? Since both gradients differ, the temperature difference depends on the wind bin. This should be accounted for.

The different slopes of the wind-temperature relationship are now discussed. The smaller slope in C21 is consistent with reduced horizontal temperature gradients. The dT is calculated using the linear relationship in C20 (i.e., the blue line). This is now stated in the caption to Fig. 5. The reason for this choice is to be consistent with the methodology for separating the dynamical and thermodynamical components, in which the dynamical component assumes the wind-temperature relationship remains constant (but the wind changes) and the thermodynamical component accounts for the change in wind-temperature relationship. To further clarify, the dT is determined by taking the mean zonal wind in C20 and C21 (the two vertical lines) and finding the y-axis value where these intercept the blue line. If I correctly understand the Reviewer’s remark (“Since both gradients differ, the temperature difference depends on the wind bin”) it seems they are mistaken in thinking that dT is determined by taking a single value of x and finding two values of y , one for the intercept with the blue line and one for the intercept with the red line (with dT being the difference between these). This in fact would yield an estimate of the temperature change expected due to thermodynamics not dynamics.

Please see response below with regards to error estimation.

Actually, this is missing in the whole manuscript: error estimation. You provide two

decimals most of the time but no estimate of the error interval. The only discussion of errors / discussion of significance appear in Table 1 and 2, last column.

This is not entirely true as significance was plotted in most Figures, as well as provided in the Tables. That said, I agree that numbers quoted in the text should have uncertainty estimates provided and did not. I have gone through the text and now provide p values wherever I mention significance (or lack thereof) and added 95% confidence intervals for all quoted numbers.

Figure 5c, line 208: Something must be wrong here. For a given wind bin, HI (red) shows higher temperatures than LI (blue) according to your caption but in contrast to line 208. The red line in Figure 5d suggests that the caption for figure 5c is wrong, since ΔT must be positive and you actually state that ΔT is positive in Table 1. Please double check your data and figures!

Oops! The reviewer is correct of course, the figure caption was wrong. In Figure 5c LI was shown in red and HI in blue, not the other way round. This has been corrected.

Lines 213-216: I would object to parts of that conclusion: Especially the negative zonal wind anomalies get even cooler or stay as cool as they are. Actually, these characterize the extremes. Therefore, one has to state that the extreme negative NAO events stay as cold or become even colder. Especially the dynamical contribution is negative and is only partly cancelled out by the thermodynamical contribution. The conclusion that no cooling effect exists holds for the extreme negative NAO events of future projections (referring to figure 6f), although the dynamical effect is still negative.

The sentences referred to discuss the total effect over the full midwinter period (i.e., what is shown by the crosses on the right-hand side of Fig. 5d). I point out that the sentence starts: “Summed over all days (i.e., over all wind categories)”. The Reviewer’s remark about easterly wind categories getting colder is true and is stated explicitly in the preceding sentence: “The net contribution shows cooling on days of strong easterly flow”. I’ve edited these sentences slightly to emphasise that I am talking about the total effect across all days in NAO- midwinters.

This leads me to the question, if the temperature variability is increased? In fact, this would imply that cold extremes might be not as cold as today measured by absolute temperature, but very similar. If, simultaneously, the general variability is increased the extremes become more extreme measured by the actual average temperature, this would be an increase of extremes anyway. In terms of adaptation: We have to adapt to significantly warmer climate, but winter cold extremes stay as cold as in present climate. Thus, the conclusion significantly depends on your definition of extremes. Although it is right to claim that cold extremes do not become colder according to your data, the former aspects need some discussion!

I agree, of course, that an increase in temperature variability could lead to more (cold) extremes, even if the mean was unchanged. Or alternatively that an increase in variability could offset mean warming, leading to little change in cold extremes. As can be seen in Fig 5b (and the figure provided above), the variability of Northern

European temperature is in fact decreased due to sea ice loss. This translates in fewer cold extremes even when one accounts for the small mean warming. Decreased temperature variability in entirety consistent with past studies of the temperature response to sea ice loss (e.g. Screen, 2014; Schneider et al., 2104; Screen et al., 2015a,b; Sun et al., 2015; Blackport and Kushner, 2016). This is discussed on page 10.

Line 225-231: This whole paragraph is just a citation of ref. 64? I see only minor contributions from the present study to these results. Maybe this should be moved to conclusions or should be removed.

This paragraph described results from the present study, not ref. 64. However, the results are highly consistent with ref. 64 and in light of this, I have removed this section and Table 2.

Line 248: Fig. 3c does not exist.

Indeed, it should have said Fig. 4c. This has been corrected.

Reviewers' comments:

Reviewer #1 (Remarks to the Author):

I am satisfied by most of the revisions made by the reviewer, that help to better understand the results and support the conclusions.

I still have a concern about the impact of the reference that is chosen to define the NAO events in LI (and C21). The claim, made in l. 110-113, that the change in NAO index in LI vs HI is due to a change in variance but not in mean seems at odd with result from Fig. S1, that shows a clear NAO-signal in mean LI vs HI changes. Given this large difference in mean NAO index in the two simulations, I'm surprised that using HI as reference for the NAO index does not impact the results. With a shift in the mean distribution of the NAO index, I would expect larger NAO-anomalies in Fig. S8 compared to Fig. 1 (that uses the LI NAO mean/std dev as reference). The only explanation I see to reconcile the results here is that HI as a smaller standard deviation, and that compensates for the lower mean when selecting the NAO- events in LI based on the HI reference.

I think this question should be clarified, and I suggest to plot PDF of the NAO index in LI and HI (as well as in C20 and C21) so we see how the distributions change. I insist on this point to make sure we understand results of Fig. S8, and that the absence of cooling over Europe is still valid when choosing HI as a reference (which in my opinion is a better approach since we are more interested in extreme changes due to high vs low sea ice, rather than to changes of NAO/temperature extreme distribution with less Arctic sea ice).

Providing that this question is clarified I think that the paper is suitable for publication.

Reviewer #3 (Remarks to the Author):

The author has improved the manuscript significantly and has satisfactorily responded to every comment. The general description of the results is now more balanced as well as the description of methods is now more comprehensible. The endorsement of different statistical approaches overcomes my concerns about the robustness of results. I agree that all different approaches have their benefits and drawbacks. I am pleased to see additional information about statistical significance and confidence intervals.

I recommend the manuscript to be ready for publication in its present form.

Responses to Reviewers (Reviewer comments in black; replies in green)

Reviewer #1

I am satisfied by most of the revisions made by the reviewer, that help to better understand the results and support the conclusions.

I thank the Reviewer once more for their time and useful suggestions.

I still have a concern about the impact of the reference that is chosen to define the NAO events in LI (and C21). The claim, made in l. 110-113, that the change in NAO index in LI vs HI is due to a change in variance but not in mean seems at odd with result from Fig. S1, that shows a clear NAO- signal in mean LI vs HI changes.

On reflection, I can see how this confusion has arisen, but there is a simple explanation: the NAO index is defined at the surface (sea level pressure; SLP) whilst Fig. S1 shows the mid-tropospheric response (500 hPa; ~5km altitude). The mean response in SLP does not strongly resemble the NAO-. The text is correct in stating that there is essentially no change in the mean surface NAO index. There is however, a significant increase in its variance (an intensification of extremes). I've made a few changes to the text to make this distinction between the surface and 500 hPa clearer:

1) I've added the following sentence at line 113:

“Note that the climatological winter circulation response is NAO-like in the mid-troposphere (Supplementary Fig. 1) but not at the surface, hence no mean shift in the surface NAO index”

2) I now prefix any instance on the wording “NAO index” with the word “surface”.

3) I've added the word “mid-tropospheric” on line 65 where Supplementary Fig. 1 is referred to.

4) I've removed the text “despite a shift towards NAO-” at line 182, since the mean surface NAO does not change.

Given this large difference in mean NAO index in the two simulations, I'm surprised that using HI as reference for the NAO index does not impact the results. With a shift in the mean distribution of the NAO index, I would expect larger NAO- anomalies in Fig. S8 compared to Fig. 1 (that uses the LI NAO mean/std dev as reference). The only explanation I see to reconcile the results here is that HI as a smaller standard deviation, and that compensates for the lower mean when selecting the NAO- events in LI based on the HI reference.

I agree with this if, hypothetically, there was a large change in the mean NAO index between the simulations. However as reiterated above, the mean surface

NAO indices have equal means in the two simulations. Thus in this case, using HI or LI as a reference has little bearing on the results. The smaller standard deviation in HI means that more events are selected when using HI as a reference than when using LI as a reference; but these additional cases represent the weakest NAO- events and do not have a large influence on the composite means across all NAO- events. This can be seen in the figure below that shows the two variants of the LI NAO index (normalized relative to either LI or HI). Each NAO- event is identified by a black cross, with blue circles around events common to both indices and red circles around events unique to that index. You can see that the composites are largely formed from the same events in both cases, hence the weak sensitivity to the reference mean. I remind the Reviewer that the composite means are not themselves normalized (so irrespective of which variant of the NAO index is used, I am sampling from the same raw data) as a later comment (see below) perhaps implies confusion in this regard.

I think this question should be clarified, and I suggest to plot PDF of the NAO index in LI and HI (as well as in C20 and C21) so we see how the distributions change. I insist on this point to make sure we understand results of Fig. S8, and that the absence of cooling over Europe is still valid when choosing HI as a reference (which in my opinion is a better approach since we are more interested in extreme changes due to high vs low sea ice, rather than to changes of NAO/temperature extreme distribution with less Arctic sea ice).

For the benefit of the Reviewer I include below a plot showing the PDFs of the NAO indices (also now included in the Supplementary Information). The vertical lines show the mean (red for LI; blue for HI; note these plot on top of each other)

and 1 standard deviation below the mean. As stated in the text and reaffirmed in the responses above, the surface NAO index shows no mean change in response to sea-ice loss, but its variability is increased.

With regards to the last comment in parentheses, I think the Reviewer's line of thought is a little muddled. The normalization is only applied to the NAO index in order to select a consistent number of events. The composite means themselves are not normalized, so any mean change in say, temperature, will be present in the composites.

Providing that this question is clarified I think that the paper is suitable for publication.

The above responses and associated edits should provide the clarification sought by the Reviewer.

Reviewer #3

The author has improved the manuscript significantly and has satisfactorily responded to every comment. The general description of the results is now more balanced as well as the description of methods is now more comprehensible. The endorsement of different statistical approaches overcomes my concerns about the robustness of results. I agree that all different approaches have their benefits and drawbacks. I am pleased to see additional information about statistical significance and confidence intervals.

I recommend the manuscript to be ready for publication in its present form.

I thank the Reviewer once more for their time and useful suggestions.

REVIEWERS' COMMENTS:

Reviewer #1 (Remarks to the Author):

I acknowledge the author for the last clarifications he provided and included in the text. I believe that making clear that the NAO signal is different at the surface and at 500 hPa was important when interpreting the results.

I recommend publication for the manuscript.